# Resonance Suppression of Servo System Based on State Equalizer Method

**DOI:** 10.3390/s22176540

**Published:** 2022-08-30

**Authors:** Jinzhao Li, Yueming Song, Xiantao Li

**Affiliations:** Changchun Institute of Optics, Fine Mechanics and Physics, Chinese Academy of Sciences, Changchun 130033, China

**Keywords:** state equalizer speed closed loop, mechanical resonance, stability control

## Abstract

Aiming at the problem of mechanical resonance faced by the servo control system of the aero-optical stabilization platform, based on the proportional integral and disturbance observer combination (PI+DOB) control algorithm, a state equalizer speed closed loop is proposed. Compared with the traditional PI+DOB control algorithm, this new control structure can suppress the resonance peak and the anti-resonance peak at the same time. The experimental results show that compared with the PI+DOB control algorithm, after adding the state equalizer speed closed-loop to compensate for the model, the closed-loop bandwidth is increased by 42%. The anti-disturbance capability of the control system has been significantly improved, and it has good robustness under vibration conditions. To sum up, adding the state equalizer speed closed loop on the basis of PI+DOB has an obvious effect on the suppression of mechanical resonance and the performance improvement of the control system.

## 1. Introduction

The aviation photoelectric stabilization platform is widely used in optical reconnaissance and tracking systems in aerospace and aviation fields. Its main function is to isolate the interference of the angular movement of the carrier to the aiming device so that the sight axis of the aiming device can stare at the target stably and quickly, and reduce the image blur caused by flutter, and improve the imaging quality.

However, with the needs of modern warfare, the distance of reconnaissance is also getting farther and farther, and it has gradually moved towards ultra-long distance. As a result, the requirements for the stability and accuracy of the Los have become more and more strict. However, in the actual working process, due to airborne conditions, the weight of the aviation photoelectric stabilization platform is always strictly limited. At the same time, the cost must be taken into account. The low frequency of the material selection structure of the platform causes the mechanical resonance frequency of the platform to be too low which directly affects the design of the servo controller. Figure 1 shows several representative airborne aviation photoelectric stabilization platforms.

In the traditional controller design process, not only should the crossover frequency of the open-loop curve of the system be ensured, but also a sufficient mid-frequency bandwidth should be ensured. However, the open-loop shear frequency of the system is tightly controlled by the mechanical resonance frequency. As a result, it is difficult to further increase the bandwidth of the system controller, which limits the anti-disturbance capability of the aviation optoelectronic stabilization platform. In the advanced control process, such as the sliding mode control process, due to the effect of mechanical resonance, the switching discontinuity of itself is more obvious. The chattering phenomenon will be more severe, which limits the stability of the control system. In the design process of the active disturbance rejection controller, when there is mechanical resonance combined with the mechanical resonance with a higher peak value at the resonance frequency point, the amplitude margin is reserved to ensure the stability of the system. As a result, in a steady state, the open-loop cut-off frequency of the system needs to be smaller than the structural resonance frequency of the controlled object, which limits the further improvement of the system bandwidth [1,2,3]. While the stability of the extended state observer is related to the observer bandwidth, the stability of the control closed-loop system is related to both the observer bandwidth and the controller bandwidth. Moreover, the stability domain of the extended state observer and control closed-loop system increases monotonically with the increase of the corresponding bandwidth. Therefore, the existence of mechanical resonance will reduce the inhibitory effect of ADRC on external disturbances [4,5].

To sum up, the mechanical resonance of the aviation photoelectric stabilization platform has a great limiting effect on the bandwidth of the servo control system, which in turn affects the design of the controller, and it is difficult to ensure that the control system has a satisfactory stabilization effect. Therefore, many methods for suppressing mechanical resonance have emerged, for example, by changing the structural form to reduce the moment of inertia of the load or changing the quality of the structural material by using a more rigid material to increase the peak frequency of mechanical resonance. The cost of this method is relatively high, and the effect of improving the mechanical resonance is not obvious. It can only increase the mechanical resonance frequency to a certain extent and cannot directly suppress or solve the mechanical resonance [6,7,8,9]. The improvement of the mechanical resonance frequency can also be achieved by designing the controller. Since the servo drive system can directly detect the electrical position and speed by using the encoder, once the control algorithm involves other state quantities, such as shaft torque, load torque, and load position, it must involve the design of the observer [10]. The PI control combined with the state feedback method theoretically increases the freedom of the system poles, but the design of the observer is too complicated [11,12]. In addition, there is also the problem that the PID controller has a poor control effect when dealing with the system with parameter uncertainty, nonlinearity and external disturbance [13,14,15,16]. For this reason, a corresponding adjustment technology is proposed to improve the robustness of the system, but this method requires too much system hardware in practical applications [17]. Then, a parameter design method of frequency-domain PID regulator is proposed to adapt it to the application of time-varying load inertia [18]. However, compared with the genetic algorithm parameter tuning technique, the frequency domain method introduces a large dynamic overshoot. On the premise of ignoring transmission and load characteristics, an explicit model predictive controller can be designed, and the safety limit of electromagnetic torque and transmission shaft torque during system operation is considered in the control algorithm [19]. Based on the realization of axial torque limit control by Model predictive control (MPC), its potential in position tracking of aero-optical stabilization platform is studied [20]. When the moment of inertia on the load side changes, the matrix parameters in the MPC objective function should be appropriately changed to meet the new system dynamic characteristics; however, it has a large amount of calculation and is difficult to control easily. For the mechanical resonance of the flexible frame of the aero-optical stabilization platform, an additional feedback control method can be performed by using the speed difference between the motor and the load [21,22,23,24]. The experimental results show that adding an additional feedback signal to the feedback channel of the speed loop in the motor can significantly reduce the resonance phenomenon of the system [25]. However, there is a problem that the full-state feedback controller is unstable. The relative speed of the drive and the load can be used as a state variable to design the full-state feedback controller to improve the tracking accuracy of the position [26]. The full state feedback controller can weaken the mechanical resonance generated during the vibration test, but it cannot guarantee the stability and robustness of the system. Adaptive neuro-fuzzy sliding-mode control, as well as fractional-order disturbance observer and single-neuron-based fuzzy PI control can be exploited when using an improved sliding-mode controller with an integral function and a fuzzy gain value to suppress the mechanical resonance of an elastically coupled system device and other methods [27,28]. A single neuron controller is embedded in each unit of the fuzzy control table, making the fuzzy control an adaptive nonlinear controller. However, because the algorithm is too complex, computationally intensive and time-consuming, it cannot be widely used in current chips [29].

Combined with the shortcomings of the above methods of suppressing mechanical resonance, this paper proposes a state equalizer speed closed-loop technology to suppress the disturbance of mechanical resonance to the Los of the aviation optoelectronic stabilization platform. The method has the characteristics of simple structure, accurate and rapid suppression of resonance and has high practical engineering application value.

## 2. State Equalizer Velocity Closed Loop Design

In this section, a simplified model of the aero-optical stabilization platform is firstly designed to illustrate the influence of mechanical resonance on the output torque of the actuator, which leads to the instability of the control system of the aero-optical platform. Secondly, the influence of mechanical resonance on the disturbance observer algorithm in the servo control system is explained; finally, on the basis of the disturbance observer algorithm, the state equalizer speed closed-loop loop is added to the control system, and the transfer function of the new closed-loop control system is analyzed. It can effectively eliminate the influence of mechanical resonance on the bandwidth of the aviation optoelectronic platform controller.

### 2.1. Mechanical Resonance Analysis of Servo System of Aviation Photoelectric Stabilization Platform

The servo system of the aviation photoelectric stabilization platform includes three loops, which are: the position loop, speed loop and current loop from outside to inside. The position loop is mainly used to track the target, and the position here refers to the relative position between the camera and the carrier, not the inertial space position. The stability of the Los is mainly realized by the speed loop. The speed loop itself has the function of keeping the Los stable in the inertial space, and most of the anti-disturbance algorithms also use the speed loop to play a role. This is used to precisely reproduce or follow a process. The design of the current loop is mainly aimed at the influence of the motor back electromotive force on the control current. Since it does not involve motion control, it will not be affected by mechanical resonance. Therefore, its control bandwidth can be very high.

The actual aviation optoelectronic stabilization platform system generally includes two frames, azimuth and pitch, and each frame and the motor are elastically connected. Figure 2 shows the elastic position distribution of the aviation photoelectric stabilization platform. There are elastic deformations between the shaft system, such as the transmission shaft and the connecting shaft and the frame. As a result, there is a resonance point in the system, and mechanical resonance is induced, which in turn affects the further improvement of the bandwidth of the velocity loop, resulting in a decrease in the stability of the Los of the platform. For the purpose of research, the servo control system of the aviation photoelectric stabilization platform can be represented by a simplified system as shown in Figure 3. Taking the azimuth axis of the aviation photoelectric stabilization platform as an example, the azimuth frame is composed of two parts: the moment of inertia of the turntable base and the moment of inertia of the load. It corresponds to the simplified system in Figure 3, where $J_{M}$ is the moment of inertia of the motor and $J_{L}$ is the moment of inertia of the motor load. The elastic impedance of an elastic material consists of the viscous damper coefficient b and the stiffness coefficient *K*. $K_{R}$ is a measuring device, such as a gyro, $K_{A}$ is an amplifier, and $K_{T}$ is the torque constant of the motor. In the aviation optoelectronic stabilization platform system, the elastic impedance is the key factor affecting the resonance response characteristics and the anti-resonance coefficient *Q*.

The impedance of the simplified system in Figure 3 is defined as a control system that takes the motor speed ${\dot{\theta }}_{M}$ as input and outputs the motor side torque *b*. Then at point a, the transfer function of the motor load impedance can be obtained as:(1)$$\frac{{\dot{\theta }}_{M}}{T}=M_{\left(s\right)}=\frac{1}{(J_{M}+J_{L})sF}$$

Among them, *F* is the resonance peak factor, and *F* can be expressed by the following formula:(2)$$F=\frac{s^{2}/\omega_{R}^{2}+(b/k)s+1}{s^{2}/\omega_{AR}^{2}+(b/k)s+1}$$

The anti-resonant frequency $\omega_{AR}$ and the resonant frequency $\omega_{R}$ in Formula (2) are expressed by Formulas (3) and (4), respectively:(3)$$\omega_{AR}={\left(\frac{K}{J_{L}}\right)}^{1/2}$$
(4)$$\omega_{R}={\left[\frac{K}{\left(J_{M}J_{L})(J_{M}+J_{L}\right)}\right]}^{1/2}$$

In Formula (3), $\omega_{AR}$ is generally defined as the anti-mechanical resonance frequency, and the anti-mechanical resonance frequency is only related to the elastic constant and the load inertia. The mechanical resonance frequency $\omega_{R}$ is related to the elastic constant, the moment of inertia of the motor, and the moment of inertia of the load. The larger the value of the elastic coefficient *K*, the higher the mechanical resonance frequency, and the higher the allowable frequency bandwidth of the servo system of the aero-optical stabilization platform. It is also less prone to mechanical resonance, and the value of $\omega_{R}$ is always greater than the value of $\omega_{AR}$. Since the damping term coefficients of $\omega_{R}$ and $\omega_{AR}$ are the same, when Equation (2) is a standard quadratic form, it has the same damping ratio, and its form is as follows:(5)$$\frac{b}{k}=\frac{2\zeta_{R}}{\omega_{R}}=\frac{2\zeta_{AR}}{\omega_{AR}}$$

From Equation (5), the resonance damping ratio coefficient can be obtained as shown in Equation (6):(6)$$\zeta_{R}=\frac{\omega_{R}}{\omega_{AR}}\zeta_{AR}$$

It can be known from Equation (6) that the amplitude of the resonance peak point is more attenuated than the amplitude of the anti-resonance peak point. From the block diagram of the closed-loop control system in Figure 4, the transfer function can be calculated as:(7)$$\frac{{\dot{\theta }}_{M}}{E_{C}}=\frac{K_{CL}}{F(s/\omega_{M})+1}$$

In the formula, the resonant closed-loop gain is $K_{CL}=K_{A}K_{M}/(1+K_{A}K_{M}K_{P})$, the cut-off frequency of the closed-loop motor control system is $\omega_{M}=\omega_{m}(1+K_{A}K_{M}K_{P})$, and the cutoff frequency of the open loop control system is $\omega_{M}=\frac{K_{e}K_{T}}{R(J_{L}+J_{M})}$.

The structural weight reduction of the aviation optoelectronic stabilization platform and the structural stiffness, elastic coefficient and moment of inertia that affects the mechanical resonance are taken into account. The inner frame is generally made of titanium material, and the rest of the outer frame is made of aluminum material. The performance parameters of the material are shown in Table 1. It can be seen from the specific stiffness parameters in Table 1 that in an ideal aero-optical stabilization platform control system, it can be considered that when the stiffness coefficient *K* is close to infinity, the value of *F* is close to 1. When *F* = 1, Equation (7) represents the closed-loop servo control system of the aviation optoelectronic stabilized platform that is not affected by resonance.

The intermediate variable of the aviation photoelectric stabilization platform control system is the motor speed ${\dot{\theta }}_{M}$, and the output variable is the motor torque *T*, which is a function of the current *I*. The influence of mechanical resonance on the velocity variable can be reflected in the Bode diagram of Equation (7). The influence of mechanical resonance on the output torque is the key to understanding the phenomenon of mechanical resonance and even the instability of the control system of the aero-optical stabilization platform.

As shown in Figure 4, the current response in the platform mechanically resonant closed loop is:(8)$$\frac{I}{E_{C}}=\frac{(K_{A}/R)s}{s+\omega_{M}/F}$$

Using Equations (7) and (8), the relationship between the two dynamic variables can be established. In the range of the anti-resonance frequency, the difference between the dynamic response of the motor speed and the current is the basis for the design of the state equalizer.

According to Equations (7) and (8), the frequency response Bode diagrams of the motor speed function and the current function are drawn as shown in Figure 5. At the anti-resonant frequency $\omega_{AR}$, the speed of the motor drops a lot and is 180° out of phase with the input $E_{C}$. On the other hand, the current gain rises rapidly to a maximum value at $\omega_{AR}$ and is in phase with the input $E_{C}$. That is, when the rate is the smallest, within the range of $\omega_{AR}$, the amplitude of the current is always the largest. The influence of the motor backlash or hysteresis on the resonant frequency of the aviation photoelectric stabilization platform is also reflected in the amplitude change of the control voltage *E* and has an effect within the range of the anti-resonance frequency change. This results in that although there is an anti-resonance point near the resonant frequency of the platform, it can suppress the amplitude of the resonant point frequency to a certain extent, but the two frequencies have a certain distance, and the suppression effect is not obvious. To sum up, when there is nonlinear interference (such as the flexible connection between the motor and the frame), the linear compensation technology cannot effectively overcome the mechanical resonance limitation of the aero-optical stabilization platform.

From the damping ratio Formula (5) and the anti-resonance frequency Expression (3), the formula for the anti-resonance damping ratio of the aviation optoelectronic stabilized platform can be obtained as follows.
(9)$$\zeta_{AR}=\frac{1}{2}\frac{b}{{\left(KJ_{L}\right)}^{1/2}}$$

In the system of a general aviation optoelectronic stabilization platform, the stiffness coefficient *K* and the load inertia $J_{L}$ are both known. If the coefficient *b* of the viscous damper can be analyzed in the frequency response characteristic, the time domain expression of the resonance peak can be expressed. The following is a method to obtain the approximate viscous damper coefficient *b*.

The resonance depth parameter is the most commonly used method to express the difference between the two frequency points before and after the amplitude attenuation to −3 dB. Moreover, considering that the central anti-resonance frequency is $\omega_{AR}$, the anti-resonance coefficient $Q_{AR}$ can be expressed by the following formula:(10)$$Q_{AR}=\frac{\omega_{AR}}{\omega_{2}-\omega_{1}}$$

The anti-resonance coefficient can also be represented by the viscous damper coefficient *b* and the stiffness coefficient *K* in the parallel system, and its specific form is as follows:(11)$$Q_{AR}=\frac{{\left(KJ_{L}\right)}^{1/2}}{b}$$

From Equations (9) and (11), it can be known that the anti-resonance damping ratio is:(12)$$\zeta_{AR}=1/2Q_{AR}$$

Then the effect of the viscous damper coefficient b on the anti-resonance peak *Q* is shown in Figure 6. By calculating the amplitude of the motor speed Bode diagram in Figure 6, when the amplitude of the motor speed attenuates to −3 dB, the difference between the two frequency points is $\left(\omega_{2}-\omega_{1}\right)$. The value of $Q_{AR}$ can be calculated to obtain an approximation of the viscous damper coefficient *b*. It can be analyzed from Figure 6 that the larger the viscous damping coefficient, the smaller the corresponding anti-resonance peak value will be.

The azimuth axis of the aviation photoelectric stabilization platform is further modeled by white noise sweep frequency, and then the ident toolbox of MATLAB is used for data processing. The amplitude-frequency characteristic curve and phase-frequency characteristic curve of the structure model of the aviation photoelectric stabilization platform is obtained as shown in Figure 7.

It can be seen from the response curve of the platform model that there is a mechanical resonance in the model, where 300.5 rad/s is the anti-resonance point of the platform structure, and 431.1 rad/s is the resonance point of the platform structure. At the same time, the phase of the model also changes abruptly between the two resonance peaks, as shown in Figure 7. The above two resonance peaks have a great impact on the closed-loop control performance of the system. First, the existence of mechanical resonance will lead to inaccurate model identification, which in turn leads to the inaccurate design of subsequent controllers. Secondly, in closed-loop control, the sudden change of system gain and phase near the resonance point and anti-resonance point makes the system prone to chattering or even instability. Therefore, it is of great significance to use an appropriate method to suppress the influence of the disturbance caused by the mechanical resonance on the design of the control system.

### 2.2. Disturbance Observer

Disturbance Observer (DOB) is a control method designed according to the principle of the internal model, and it is also one of the most commonly used anti-disturbance control algorithms. The basic principle is to estimate the difference between the actual output and the ideal output of the controller through the nominal model and use it as an estimated disturbance to compensate for the control quantity, thereby achieving the purpose of suppressing external disturbances. The basic schematic diagram is shown in Figure 8, where $G_{p}{}_{\left(s\right)}$ is the actual model of the controlled object; $G_{pn}{}_{\left(s\right)}$ is the nominal model of the controlled object; $Q_{\left(s\right)}$ is the low-pass filter. It can reduce the influence of measurement noise on the system stability, but it will cause the phase decay of the disturbance observation value, and then affect the compensation effect of the disturbance; d is the total external equivalent disturbance; $\hat{d}$ is the disturbance amount observed by the DOB; ξ is the detection noise; u is the input of the control system; θ is the output of the control system.

According to the above Figure 8, the relationship between the total output θ of the system, the input u and the equivalent disturbance *d* can be deduced at this time, as shown in Equation (13), $G_{u\theta (s)}$ is the transfer function from the control quantity to the system output, $G_{d\theta (s)}$ is the transfer function from the equivalent disturbance to the system output, and $G_{\xi \theta (s)}$ is the transfer function from the measurement noise to the system output:(13)$$\theta_{\left(s\right)}=G_{u\theta (s)}u_{\left(s\right)}+G_{d\theta (s)}d_{\left(s\right)}+G_{\xi \theta (s)}\xi_{\left(s\right)}$$

In:$$G_{u\theta (s)}=\frac{G_{pn(s)}G_{p(s)}}{G_{pn(s)}+(G_{p(s)}-G_{pn(s)})Q_{\left(s\right)}}\;G_{d\theta (s)}=\frac{G_{pn(s)}G_{p(s)}(1-Q_{\left(s\right)})}{G_{pn(s)}+(G_{p(s)}-G_{pn(s)})Q_{\left(s\right)}}\;G_{\xi \theta (s)}=\frac{G_{p(s)}Q_{\left(s\right)}}{G_{pn(s)}+(G_{p(s)}-G_{pn(s)})Q_{\left(s\right)}}$$

In an ideal situation, the nominal model of the plant is equal to the actual model $G_{p(s)}=G_{pn(s)}$, and the low-pass filter $Q_{\left(s\right)}$ has a gain of 1 at low frequencies at this time:(14)$$G_{u\theta (s)}\approx G_{pn(s)},G_{d\theta (s)}\approx 0,G_{\xi \theta (s)}\approx 1$$

It can be seen from (14) that when the disturbance *d* has no effect on the output, the external equivalent disturbance of the system is completely suppressed, but the sampling noise is also added to the output through the filter without limitation. At the same time, in order to suppress the influence of the high frequency noise of the sensor, the gain of $Q_{\left(s\right)}$ in the high frequency band should be 0. If the high-frequency components in the disturbance d are to be suppressed, the cutoff frequency of the low-pass filter $Q_{\left(s\right)}$ is required to be as high as possible; but at the same time, in order to suppress the high-frequency noise caused by the mechanical resonance of the aero-optical stabilization platform structure, the cut-off frequency of $Q_{\left(s\right)}$ is required to be as low as possible, which is the biggest contradiction faced by the interference observer in the application process. In this paper, the speed closed-loop method of the state equalizer can be used to increase the frequency of the mechanical resonance of the aviation photoelectric stabilization platform, The cutoff frequency of a is further increased to take into account the robust stability of the interference observer and the ability to suppress high-frequency noise. In the actual working process, due to the existence of mechanical resonance, the design of $Q_{\left(s\right)}$ is limited, which in turn limits the further improvement of the system disturbance suppression capability. For the high-frequency mechanical resonance frequency, since it is much larger than the disturbance frequency of the system, it has little effect on the design of $Q_{\left(s\right)}$ and can be ignored. For the low-frequency mechanical resonance frequency, its impact on the design of $Q_{\left(s\right)}$ how to overcome the limitation of the low-frequency mechanical resonance frequency is also the focus of this paper. In the DOB algorithm, the choice of the low-pass filter $Q_{\left(s\right)}$ has a certain influence on the phase of the control system. Although the effect of the low-pass filter $Q_{\left(s\right)}$ on the phase cannot be completely eliminated by adding a state equalization speed closed loop, the red curve (PI+DOB + state equalizer) and the blue curve (PI+DOB) are shown in Figure 9. As can be seen from the figure, the speed closed-loop method of the state equalizer proposed in this paper can improve the system bandwidth, achieve the purpose of suppressing the system resonance and improve the system anti-interference ability. Moreover, on this basis, the bandwidth of $Q_{\left(s\right)}$ can be designed to be higher. However, how to design a higher $Q_{\left(s\right)}$ bandwidth and reduce the impact of phase lag on the system’s anti-jamming capability is also the focus of future research in this paper.

### 2.3. State Equalizer Speed Closed Loop

When the state equalizer speed closed-loop is applied to the motor speed and system current response characteristics in Figure 4, a balanced speed response will be produced, regardless of the magnitude of the input signal, the motor speed and torque response characteristics will remain balanced over the resonant frequency range. Finally, the frequency response of the system is made smooth, which has a good inhibitory effect on the mechanical resonance of the control system of the aviation photoelectric stabilization platform.

The design method of the state equalizer is to add a resonant equalization speed closed loop based on the closed-loop resonant circuit of Figure 4, as shown in Figure 11, to suppress the influence of mechanical resonance on the stability of the optoelectronic control system. After the design of the state equalizer is completed, it can be reversely compensated into the control system to achieve the purpose of correcting the model and suppressing the resonance. The schematic diagram of its model calibration is shown in Figure 10.

In order to obtain the speed and current closed-loop response characteristics of the platform control system, the function of the armature current I (torque) is subtracted from the voltage signal $E_{R}$ (speed) of the measurement data of the gyro and other measuring instruments. According to Figure 11, the closed-loop rate transfer function of the following state equalizer can be obtained as follows:(15)$$\frac{{\dot{\theta }}_{M}}{E_{C}}=\frac{K_{A}K_{M}G_{A}}{\left(s/\omega_{m})F[1-(\gamma K_{\gamma }H)(K_{A}G_{A}/R)]+(1+K_{A}K_{M}K_{R}G_{A}\right)}$$

Comparing Equation (7) with Equation (15), it can be seen that the resonance crest factor *F* is multiplied by the denominator term of the term $K_{r}H$ including the equalizer. At this time, the parameter term of the state equalizer can be expressed as the following equation.
(16)$$K_{\gamma }H=\frac{R}{K_{A}G_{A}}$$

And: the coefficients of the state equalizer γ=1.

The closed-loop transfer function can be simplified to:(17)$$\frac{{\dot{\theta }}_{M}}{E_{C}}=\frac{K_{A}K_{M}G_{A}}{1+K_{A}K_{M}K_{R}G_{A}}$$

It can be seen from Figure 10 that the value of the state balance coefficient γ in Equation (17) should not exceed 1, which can avoid the instability of the current closed-loop feedback loop, and its value range can be between 0–1, depending on the actual mechanical resonance frequency value. When γ is equal to 1, or close to 1, the mechanical resonance frequency $\omega_{m}$ in Equation (15) is also eliminated. At this time, the closed-loop bandwidth of the aviation photoelectric stabilization platform control system is mainly determined by the amplifier coefficient $G_{A}$. At this time, the resonance crest factor *F* can be completely eliminated, and then the influence of mechanical resonance on the control system of the aviation photoelectric stabilization platform can be eliminated. It can further improve the bandwidth of the speed loop servo controller of the aviation photoelectric stabilization platform, so as to achieve the purpose of suppressing the influence of external disturbances on the platform. Ultimately, the Los stabilization accuracy of the aviation photoelectric stabilization platform is higher.

## 3. Experimental Validation and Data Analysis

The experiment is carried out on a two-axis four-frame aviation photoelectric stabilization platform driven by a certain type of inner frame brushless motor. The experimental platform is shown in Figure 12. The experiment is carried out in the azimuth axis of the aviation photoelectric stabilization platform. In order to conduct a comprehensive test of the performance of the compensation algorithm of PI+DOB using the speed closed loop of the state equalizer, in this paper, the bandwidth test experiment, the disturbance rejection ability experiment, and the adaptability experiment of the speed closed loop of the equalizer in different states using the optoelectronic platform structure environment are carried out, respectively. As a comparison, the PI+DOB disturbance compensation control algorithm has also been carried out in the above experiments.

### 3.1. Bandwidth Test

After using the state equalizer speed closed loop to compensate the control system of the aviation photoelectric stabilization platform, The following tests the bandwidth of PI+DOB and PI+DOB+ State equalizer speed closed loop. After the bandwidth test on the azimuth axis, the comparison chart of the amplitude-frequency characteristics of the bandwidth of the two control algorithms is shown in Figure 13, and the performance comparison before and after compensation is shown in Table 2.

It can be seen from Figure 13 and Table 2 that the closed-loop bandwidth of the PI+DOB+ state equalizer algorithm (red) can reach 43.08 Hz, which is 42% higher than the controller bandwidth of the aviation photoelectric stabilization platform using the PI+DOB control algorithm (yellow) alone. It shows that after adding the state equalizer speed closed loop in the controller, it can inhibit the machinery of the optoelectronic platform. The bandwidth of the closed-loop control system is significantly improved, and it fully meets the needs of the aviation optoelectronic stabilization platform in practical work.

### 3.2. Disturbance Suppression Ability Experiment

#### 3.2.1. Velocity Stability Experiment

In order to compare the ability of the state equalizer algorithm to suppress the external disturbance caused by mechanical resonance based on the PI+DOB control algorithm of the aviation photoelectric stabilization platform, the experimental aviation photoelectric stabilization platform is installed on a high-frequency five-axis flight table to simulate the flight state. The specific installation results are shown in Figure 14:

In the experiment, the platform is set at the zero position, and the expected rotation speed of the aviation photoelectric stabilization platform is set to zero, and then the isolation ability of the platform speed loop to the disturbance is judged by measuring the gyro values. Figure 15 shows the speed stability when the flight table moves sinusoidally at 1° and 2 Hz, and the photoelectric stabilization platform adopts the PI+DOB controller and the PI+DOB+ state equalizer controller, as well as the static error of the gyro noise.

Obviously, compared with the PI+DOB controller, after adopting the state equalizer speed closed-loop, the amplitude and duration of the platform speed peak are significantly reduced, and its peak value is about 0.04 °/s. Considering that the noise of the gyro is relatively large, the peak value of the gyro noise when the platform is absolutely stationary is 0.02 °/s, this shows that in such a noisy system, the boost limit of this system is 0.02 °/s, It can be seen that the photoelectric stabilization platform with the PI+DOB control algorithm of the state equalizer speed closed-loop control algorithm has very satisfactory results in suppressing disturbance.

After performing spectrum analysis on the data of (a) and (b) in Figure 15, the results are shown in Figure 16 (orange is the data spectrum analysis of the state equalizer speed closed loop on the basis of PI+DOB, and the green is the data spectrum analysis of using PI+ Data spectrum analysis of DOB controller). It can be clearly seen from the comparison that in the control system that adds the state equalizer speed closed loop based on PI+DOB, the residual amount of disturbance at 2 Hz is about 1/4 of the photoelectric platform using the PI+DOB controller alone, that is, the speed disturbance isolation is improved by 12.04 dB.

#### 3.2.2. Target Tracking Experiment

In order to make the frame of the aviation photoelectric stabilization platform move with arbitrary amplitude and frequency on the five-axis flight table, a state equalizer speed closed loop is added on the basis of the servo controller PI+DOB algorithm to maintain the stable orientation of the Los in space. In this process, the controller adding the state equalizer speed closed loop should suppress the influence of the flighting of the photoelectric platform frame, to stabilize the Los of the photoelectric platform to remain unchanged at the specified angular position. The target tracking experiment tests the closed-loop bandwidth and disturbance rejection capability of the servo control system at the same time.

The image analyzer as shown in Figure 17 analyzes and calculates the Los shaking information of the target relative to the aero-optical platform, and then compares the Los stabilization accuracy of the two-axis and four-aero-optical stabilization platforms.

Taking the situation of the flight simulation table shaking at an amplitude of 1° 2 Hz as an example, the tracker of the photoelectric stabilization platform is at the longest focal length to track an infinite fixed target. By measuring the off-target amount of the deviation of the Los relative to the target point, the stabilization accuracy of the Los of the platform is analyzed. Figure 18 shows the movement range of the target deviation from the center point when the photoelectric stabilization platform adopts the PI+DOB controller and the PI+DOB+ state equalizer controller when the flight simulation table performs sinusoidal motion at 1° 2 Hz. Comparing Figure 18a,b, it can be clearly found that the motion range of the visual axis shown in Figure 18a is significantly reduced, about ±15 urad, which is only 3/8 of Figure 18b. It shows that the control algorithm of the PI+DOB+ state equalizer controller can effectively suppress the influence of disturbance on the Los of the aero-optical stabilization platform.

### 3.3. Robustness Experiment of State Equalizer

In order to verify the stability and robustness of the closed-loop speed of the state equalizer when the mechanical resonance model changes when environmental factors, such as vibration change, the following experiments are performed to measure the stability accuracy of the Los.

In order to simulate the actual complex and changeable working environment more realistically, the specific experimental method is to install the two-axis four-frame aviation photoelectric stabilization platform system in the vibration test device. The device can carry out random vibration tests. When the mechanical resonance of the structure changes due to the structural change of the aviation photoelectric stabilization platform, the state equalizer is added on the basis of the PI+DOB controller to suppress the mechanical resonance of the optoelectronic platform. The specific operation is shown in Figure 19.

In two sets of control experiments, the frame of the aero-optical stabilization platform adopts the PI+DOB controller alone and the controller with a state equalizer speed closed-loop based on PI+DOB to control the frame stability of the aviation-optical stabilization platform.

Taking the condition that the vibration level of the shaking table is 4 ${g/\mathrm{Hz}}^{2}$ as an example, let the tracker of the photoelectric stabilization platform be at the longest focal length, and track the infinite fixed target. By measuring the off-target amount of the deviation of the Los relative to the target point, the stabilization accuracy of the Los of the platform is analyzed. Figure 20 shows the motion range of the target deviation from the center point and the off-target amount when the photoelectric stabilization platform adopts the PI+DOB controller and the PI+DOB+ state equalizer controller, respectively.

Table 3 shows the line-of-sight stabilization accuracy of the aviation optoelectronic stabilization platform under the condition of 3 ${g/\mathrm{Hz}}^{2}$ to 5 ${g/\mathrm{Hz}}^{2}$ vibration levels (the test results are also obtained by the image analysis system).

From Table 3 and Figure 20, it can be clearly seen that, compared to the Los stabilization accuracy of the test PI+DOB algorithm alone, the Los stabilization accuracy is significantly lower than the Los stabilization accuracy of the PI+DOB controller with the speed closed loop of the state equalizer. Therefore, the controller of PI+DOB with a closed-loop speed of state equalizer has strong robustness and fully meets the needs of practical engineering.

Based on the above experiments, compared with the PI+DOB control algorithm, the method proposed in this paper based on the PI+DOB control algorithm and the state equalizer speed closed-loop suppression of mechanical resonance can improve the interference suppression ability and bandwidth. Comparing the robustness of the aviation optoelectronic stabilization platform using PI+DOB alone in the controller and adding a state equalizer speed closed loop in the controller, the former is significantly lower than the latter. Therefore, the speed closed-loop control algorithm of the PI+DOB+ state equalizer fully meets the performance requirements of the optoelectronic platform in practical engineering.

## 4. Conclusions

Aiming at the mechanical resonance problem of the aviation optoelectronic stabilization platform, this paper proposes a method based on PI+DOB to add a state equalizer to the control algorithm to suppress the mechanical resonance. Compared with the current general PI+DOB control algorithm, adding the state equalizer speed closed-loop control system can better eliminate the influence of mechanical resonance on the system bandwidth. The experimental results on the aviation photoelectric stabilization platform show that the control system with the speed closed loop of the state equalizer has a good compensation effect on the mechanical resonance. After the introduction of the state equalizer speed closed-loop, the noise immunity, control bandwidth and stability accuracy of the control system have been greatly improved. Moreover, the control system that introduces the state equalizer speed closed-loop has good robustness, which can meet the practical application of aviation photoelectric stabilization platform in engineering.

At present, the control method based on PI+DOB adding a state equalizer can already meet the requirements of most control algorithms for mechanical resonance for amplitude correction. However, its phase delay is still relatively large, and this aspect still needs to be further improved, which is also the next research focus of the state equalizer.

## Figures and Tables

**Figure 1 sensors-22-06540-f001:**
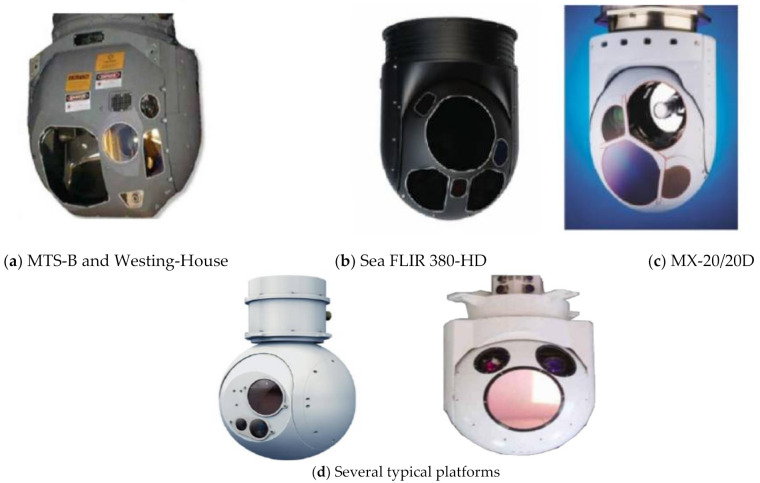
Airborne aviation photoelectric stabilization platform.

**Figure 2 sensors-22-06540-f002:**
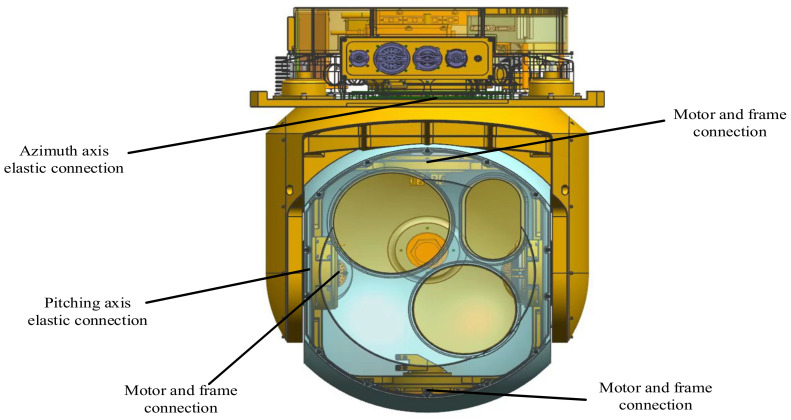
Elastic position distribution of aero-optical stabilization platform.

**Figure 3 sensors-22-06540-f003:**
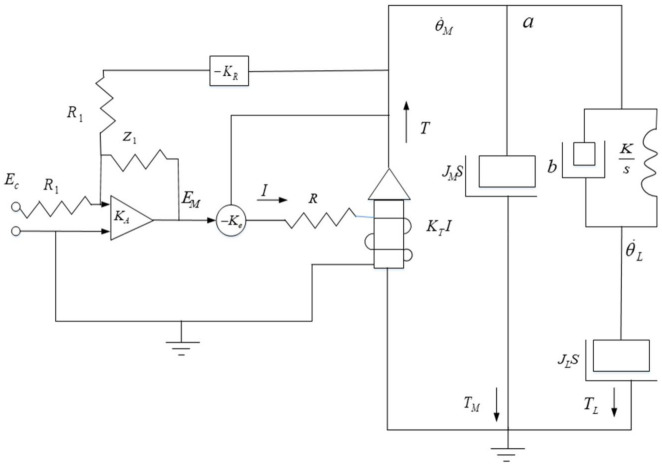
Simplified system of aviation photoelectric stabilization platform.

**Figure 4 sensors-22-06540-f004:**
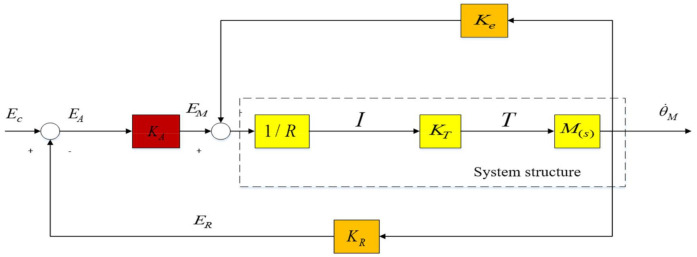
Resonant closed-loop circuit of aviation optoelectronic stabilization platform.

**Figure 5 sensors-22-06540-f005:**
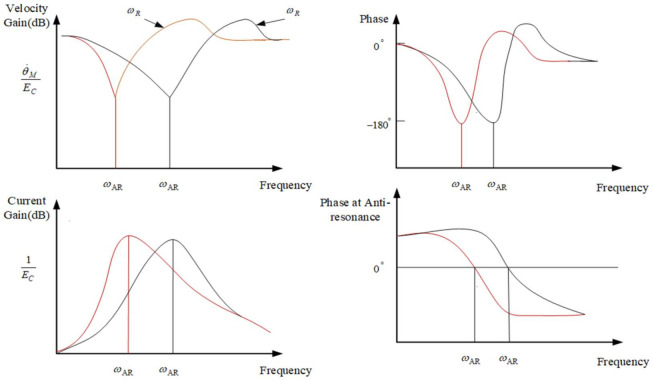
The closed-loop curve diagram of speed and current effect.

**Figure 6 sensors-22-06540-f006:**
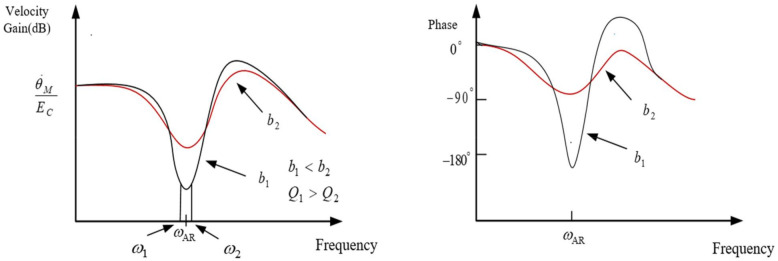
Influence of viscous damper coefficient b on anti-resonance peak *Q.*

**Figure 7 sensors-22-06540-f007:**
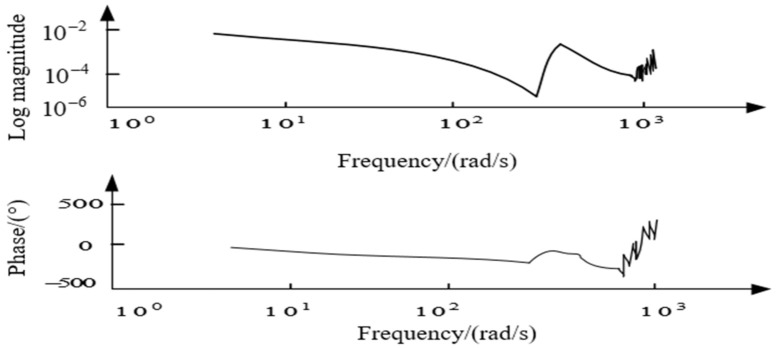
Scanning curve of platform model.

**Figure 8 sensors-22-06540-f008:**
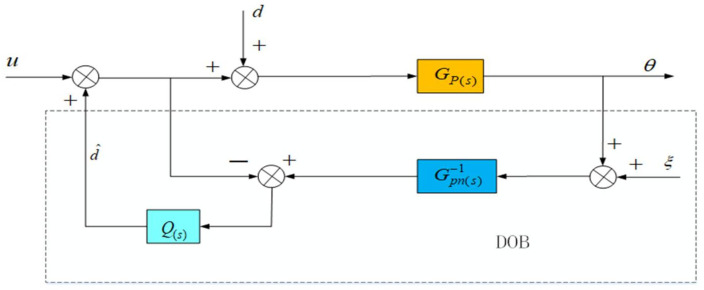
Schematic diagram of interference observer.

**Figure 9 sensors-22-06540-f009:**
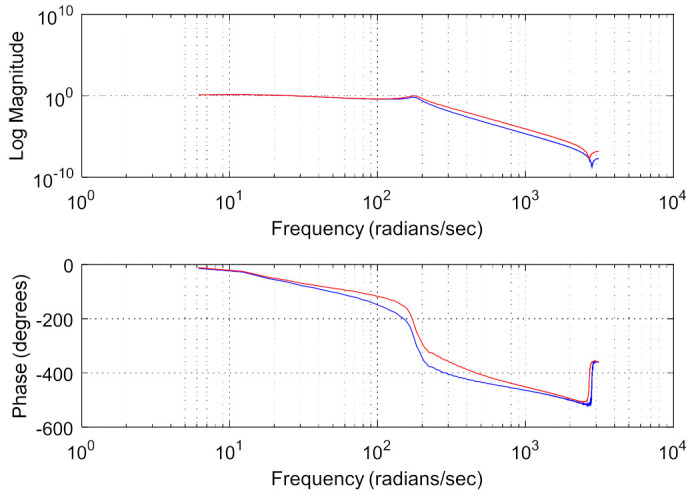
Bandwidth comparison of the two algorithms.

**Figure 10 sensors-22-06540-f010:**
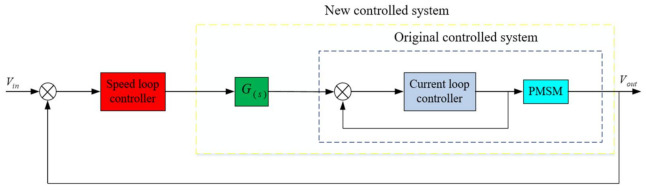
Schematic diagram of model calibration.

**Figure 11 sensors-22-06540-f011:**
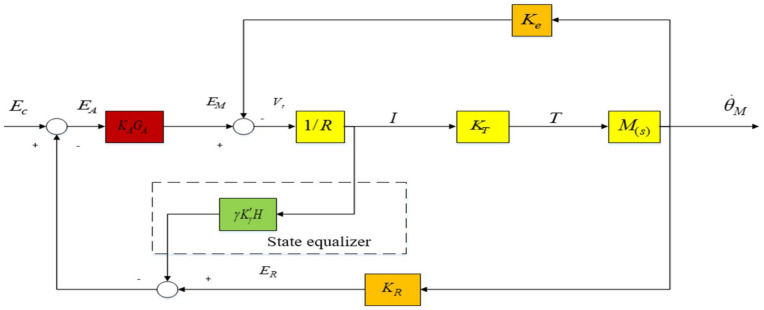
Resonant equilibrium velocity closed-loop system.

**Figure 12 sensors-22-06540-f012:**
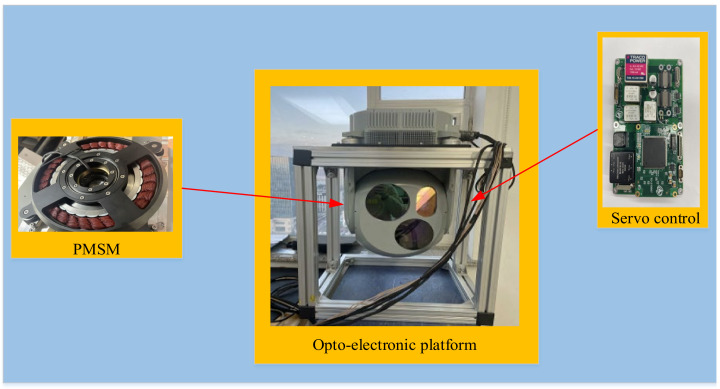
Experimental platform.

**Figure 13 sensors-22-06540-f013:**
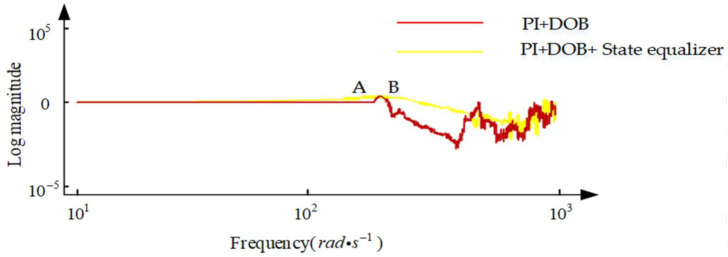
Control bandwidth comparison diagram.

**Figure 14 sensors-22-06540-f014:**
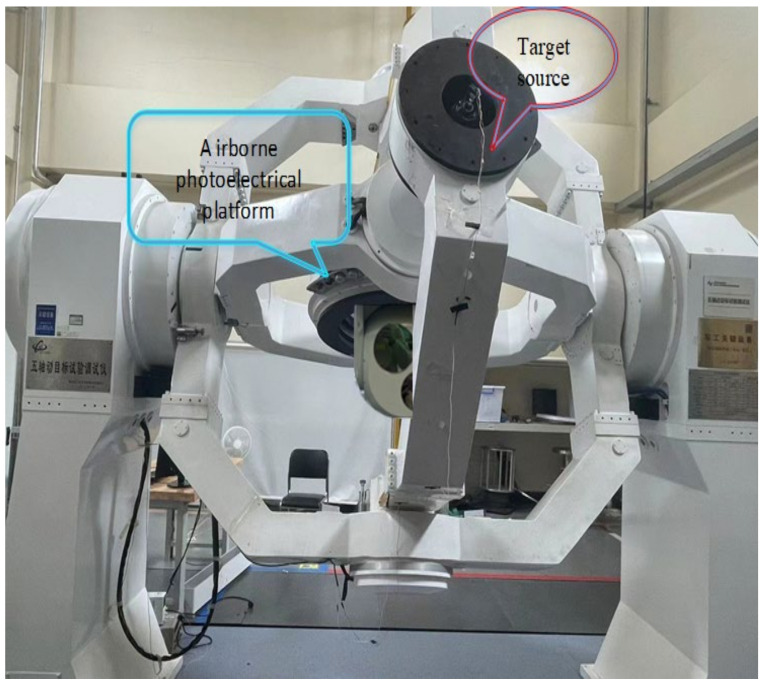
Installation diagram of photoelectric platform for anti-interference experiment.

**Figure 15 sensors-22-06540-f015:**
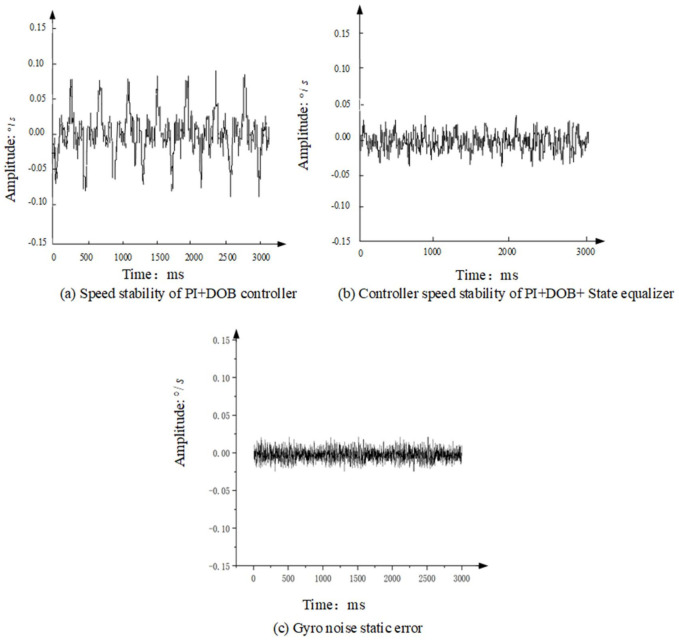
Comparison of speed stability experiments.

**Figure 16 sensors-22-06540-f016:**
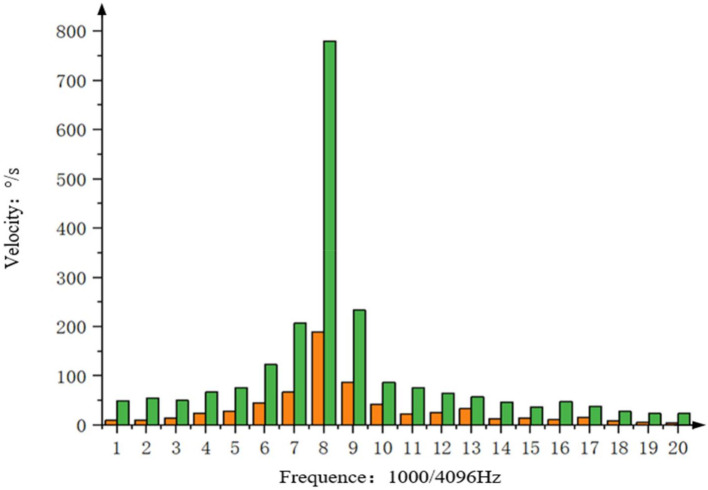
Fourier transform of the inner frame rate before and after using the state equalizer velocity closed loop under the disturbance of 2 Hz.

**Figure 17 sensors-22-06540-f017:**
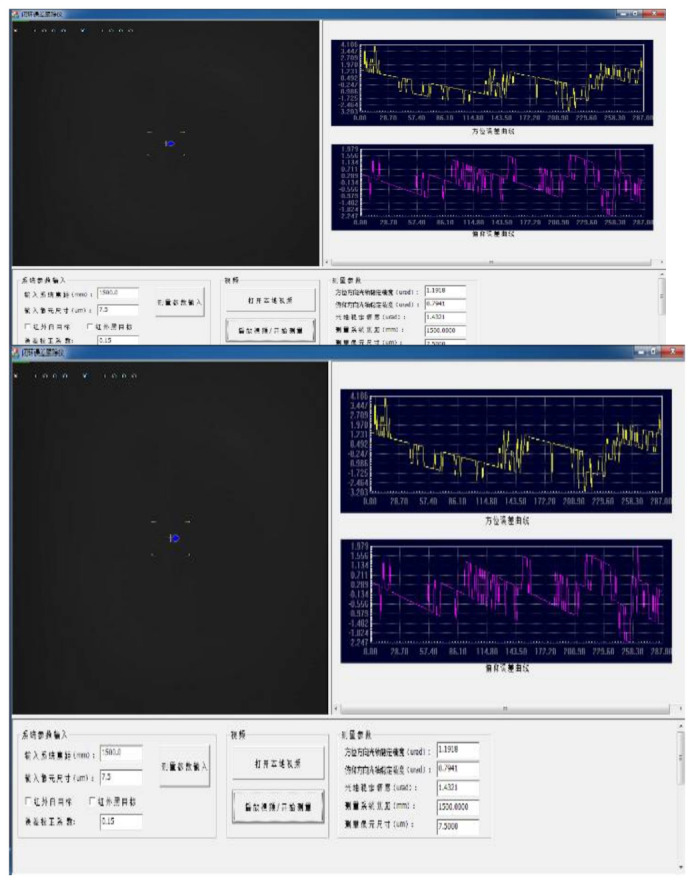
Image Analysis System.

**Figure 18 sensors-22-06540-f018:**
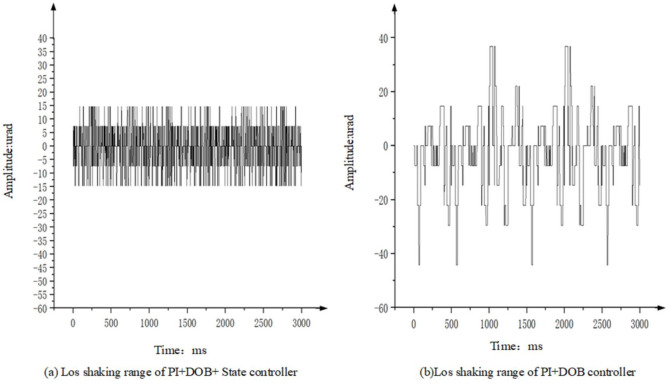
The shaking range of the Los of the controller.

**Figure 19 sensors-22-06540-f019:**
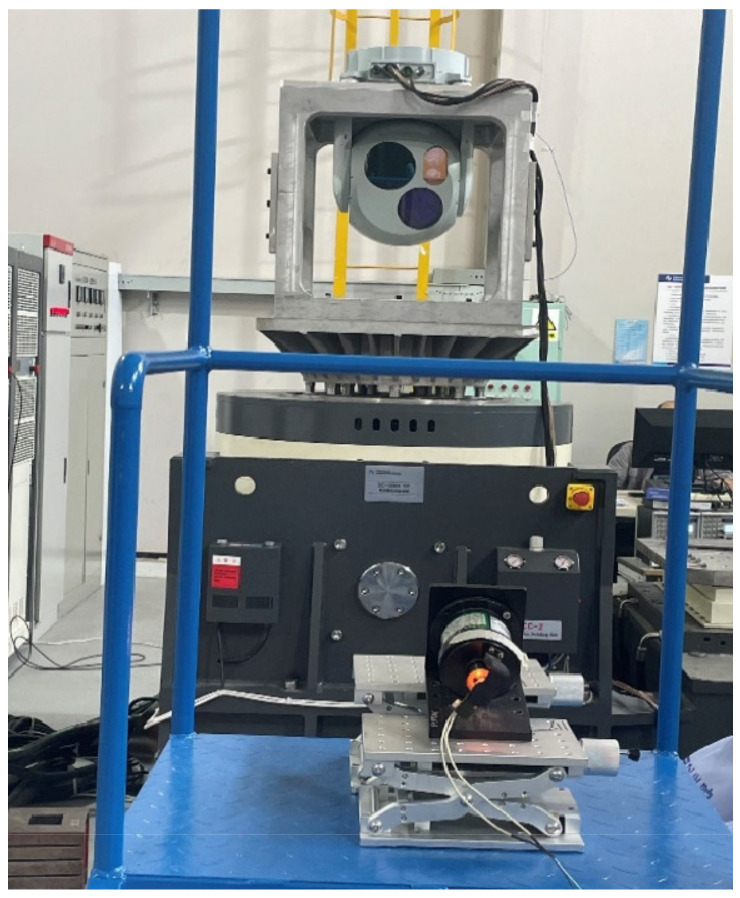
Vibration test device.

**Figure 20 sensors-22-06540-f020:**
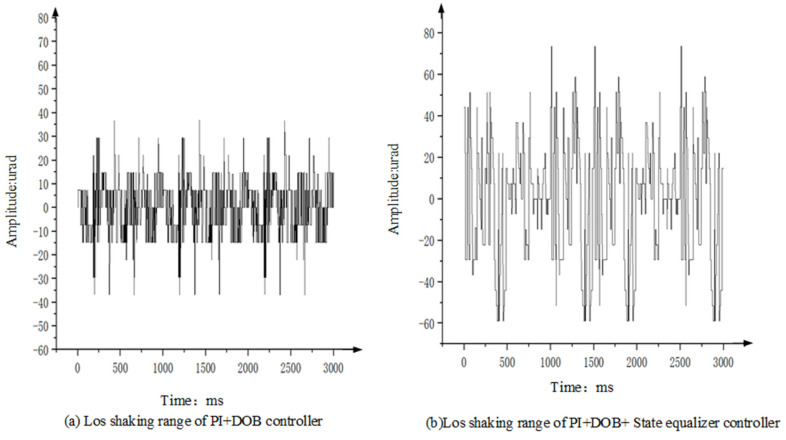
Shaking range of the controller’s Los.

**Table 1 sensors-22-06540-t001:** Common material performance parameters.

Material Name	Density $\rho (g/cm^{3})$	Tensile Strength $\delta_{b}(MPa)$	Elastic Modulus E(MPa)	Specific Strength $\delta_{b}/\rho (MN\cdot m/g)$	Specific Stiffness E/ρ(MN·m/g)
Aluminum	2.8	460	73,500	164	26,250
Titanium	4.5	941	111,720	209	24,827
Steel	7.8	1009	205,800	129.4	26,385
FRP	2.0	1039	39,200	520	19,600
Carbon fiber (high strength)	1.45	1470	137,200	1014	94,621
Carbon fiber (high model)	1.6	1049	235,200	656	147,000
boron fiber	2.1	1352	205,800	644	9800

**Table 2 sensors-22-06540-t002:** Performance comparison before and after compensation.

	PI+DOB	PID+DOB+ State Equalizer
control bandwidth	30.27 Hz	43.08 Hz

**Table 3 sensors-22-06540-t003:** Comparison of stable accuracy of the two algorithms.

Vibration Magnitude	PI+DOB	PID+DOB+ State Equalizer
3 ${g/\mathrm{Hz}}^{2}$	38.11627 µrad	13.98008 µrad
4 ${g/\mathrm{Hz}}^{2}$	45.42276 µrad	16.37992 µrad
5 ${g/\mathrm{Hz}}^{2}$	58.68746 µrad	19.87882 µrad

## Data Availability

Not application.

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
