# Peer review of "Resonance Suppression of Servo System Based on State Equalizer Method"

_sensors, 2022, doi:10.3390/s22176540_

Round 1

Reviewer 1 Report

Aiming at the mechanical resonance problem of the aviation optoelectronic stabilization platformbased on the traditional PI+DOB control algorithm, the author improves it and adds a state equalization speed closed loop. The related theoretical analysis and a large number of experiments are carried out to verify that the proposed control algorithm can effectively eliminate the influence of mechanical resonance on the system bandwidth. The overall quality of the manuscript is good and can be published with a little revision.

1)     How did the author get the actual model of the controlled object, and is there an exact expression?

2)     The author introduces a speed-related state equalizer into the proposed control algorithm, so why not carry out higher-frequency target tracking experiments, such as 10HZ 20HZ, etc.

3)     MPC on line 89 is an acronym for model predictive control. The full name should be given before using the abbreviation.

4)     The k in formula 11 should be uppercase K.

5)     The closed-loop resonant circuit on line 342 does not correspond to Figure 2 above.

6)     It is better to give the names of the experimental curves in the two colors in Figure 12 and mark the maximum bandwidth under different control methods in the Figure.

7)     The author had better give a picture of the image analyzer in Figure 16, does not have to be the operating interface of the system.

8)     Are the titles of Figures a and b in Figure 17 wrong?

Reviewer 2 Report

The article addresses the problem of mechanical resonance occurring in the control system of an aero-optical platform servo.

The paper is interesting but needs improvement at this stage.

1. The proposed method based on ma PI+DOB method introduces a relatively large delay. I would ask that the authors include another section in the article, which would include a proposal to minimize it. Please confirm the proposal by performing example simulation calculations.

2. The authors write: "... but the sampling noise is also added to the output through the filter without limitation". I would like the article to explain what the authors mean when they write about the filter without limitation.

3. In the text of the article there is a large number of inconsistently introduced markings, for example: in line 164 there is "b" and there should be "b". A similar problem occurs on lines: 165, 167, 178, 205, 206, 209, 210, 229, 244, 260, 264, 318, 357, etc.

4. There is no unit of measure on the y axis in Figure 15.

5. Figure 16 contains Chinese letters, so it is worth placing the drawing in the English language version.

6. The authors write: "By analyzing and calculating the shaking of the target relative to the visual axis by the image analyzer as shown in Figure 16." The sentence needs to be corrected.

Round 2

Reviewer 2 Report

After the changes introduced by the authors, I accept the article for publication in the mdpi journal.